# RPNet: Robust Private Inference against Malicious Clients with Adversarial Attacks

## Abstract

The increased deployment of machine learning inference in various applications has sparked privacy concerns. In response, privacy-preserving neural network (PNet) inference protocols have been created to allow parties to perform inference without revealing their sensitive data. Despite the recent advancements in the efficiency of PNet, most current methods assume a semi-honest threat model where the data owner is honest and adheres to the protocol. However, in reality, data owners can have different motivations and act in unpredictable ways, making this assumption unrealistic. To demonstrate how a malicious client can compromise the semi-honest model, we first designed a novel inference manipulation attack against a range of state-of-the-art private inference protocols. This attack allows a malicious client to modify the model output using $3\times$ to $8\times$ fewer queries relative to the current black-box attacks and accommodates larger and more complex neural networks. Driven by the insights gained from our attack, we proposed and implemented RPNet, a fortified and resilient private inference protocol that can withstand malicious clients. RPNet integrates a distinctive cryptographic protocol that bolsters security by weaving encryption-compatible noise into the logits and features of private inference, thereby efficiently warding off malicious-client attacks. Our extensive experiments on various neural networks and datasets show that RPNet achieves $\sim 91.9\%$ attack success rate reduction and increases more than $10\times$ query number required by malicious-client attacks.

## 1 Introduction

Machine-learning-as-a-service (MLaaS) is a powerful method to provide clients with intelligent services and has been widely adopted for real-world applications (Liu et al., 2017), such as image classification/segmentation for home monitoring systems (Kuna; Wyze, 2022), intrusion detection (Ashiku & Dagli, 2021), fraud detection (Raghavan & Gayar, 2019; Mishra et al., 2020). Nevertheless, the integration of MLaaS into many such applications engenders privacy concerns (Dowlin et al., 2016; Mishra et al., 2020; Lou et al., 2021). For instance, home monitoring systems like Kuna (Kuna) and Wyze (Wyze, 2022) utilize neural networks to categorize objects in user home video feeds, such as vehicles stationed near the user's residence or identifying visitors' faces, which may intrude on personal privacy.

In order to address these privacy concerns, numerous recent studies, as illustrated in Table 1, have put forth protocols for cryptographic prediction, specifically, private inference over (convolutional) neural networks, by leveraging various cryptographic primitives, e.g., fully homomorphic encryption (FHE) (Gentry, 2009). An FHE-based private inference is unique since it enables non-interactive privacy-preserving machine learning that does not require clients to present during inference. A PNet allows the user to receive the prediction outcome while simultaneously guaranteeing that neither party obtains any additional information pertaining to the user's input or the model's weight parameters. Numerous studies, for instance, Brutzkus et al. (2019); Chou et al. (2018); Hesamifard et al. (2019); Dathathri et al. (2019); Lou & Jiang (2021); Benaissa et al. (2021); Aharoni et al. (2020); Lehmkuhl et al. (2021), as indicated in Table 1, have achieved these assurances. However, all these studies make an assumption of semi-honest protocol adherence, implying that both the client-side data owner and server-side model owner comply with the protocol without malicious behaviors.

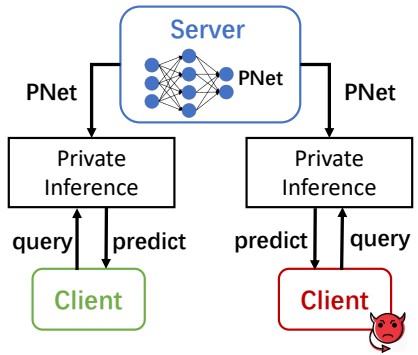

Figure 1: Client may be malicious in private inference.

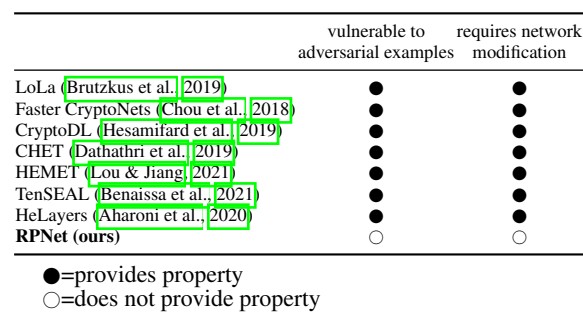

|  | vulnerable to adversarial examples | requires network modification |
|---|---|---|
| LoLa (Brutzkus et al., 2019) | ● | ● |
| Faster CryptoNets (Chou et al., 2018) | ● | ● |
| CryptoDL (Hesamifard et al., 2019) | ● | ● |
| CHET (Dathathri et al., 2019) | ● | ● |
| HEMET (Lou & Jiang, 2021) | ● | ● |
| TenSEAL (Benaissa et al., 2021) | ● | ● |
| HeLayers (Aharoni et al., 2020) | ● | ● |
| **RPNet (ours)** | ○ | ○ |

●=provides property
○=does not provide property

Table 1: Prior private inferences are vulnerable to malicious-client adversarial examples and need network modification for higher robustness.

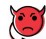

While the majority of the literature adopts the semi-honest threat model, it is fundamentally less probable that all clients will adhere to proper behavior. The server is hosted by a single service provider, and existing cloud providers employ strict access control, and physical security measures, making it considerably challenging to circumvent these safeguards. Furthermore, if a service provider is found to be acting maliciously, the repercussions could be severe due to public accountability Brasser et al. (2017); Lehmkuhl et al. (2021). In contrast, clients are numerous, operate on diverse setups under user control, have varying motives, and it suffices that one behaves maliciously as Figure 1 shows. The incentives for a client to cheat are substantial: service providers offer high-stake services such as intrusion detection, home monitoring systems, and fraud detection. Consequently, a client, who may be an attacker, could seek to obtain unauthorized access to information or manipulate the decision-making processes of the MLaaS system to attain personal or financial advantages. The recent study, MUSE (Lehmkuhl et al., 2021), investigates malicious clients attempting to pilfer models through queries, but it does not offer solutions for adversarial output manipulation attacks.

To highlight the dangers associated with a malicious client, we introduce a new adversarial output manipulation attack against semi-honest private inference protocols in Section 3. This attack enables a malicious client to modify the model's output using a reduced number of inference queries. With an efficiency improvement of $3\times \sim 8\times$ compared to the most effective prediction manipulation attacks for plaintext inference (Guo et al., 2019; Andriushchenko et al., 2020), this attack demonstrates that the utilization of semi-honest private inference protocols can significantly enhance a malicious client's ability to alter model predictions.

To counteract such amplification of security risk, we introduce RPNet, a robust private inference protocol designed to function effectively under a client-malicious threat model. In this context, the server is presumed to maintain semi-honest behavior, while the client may diverge substantially from the protocol's stipulations. As we will elaborate in Section 4, adopting this model empowers RPNet to surpass current state-of-the-art methods in its performance efficiency.

**Contributions.** We summarize our contributions as follows: (i) We design a novel inference manipulation attack, denoted by *PNet-Attack*, against private inference protocols that rely on fully homomorphic encryption. This attack enables a malicious client to manipulate the model's output with $3\times \sim 8\times$ fewer queries than the state-of-the-art. (ii) We introduce *RPNet*, a robust private inference protocol resilient to malicious clients. In designing *RPNet*, we propose to add cryptography-compatible noise in the features and logits layer. In addition, we introduce a dynamic noise training (DNT) technique to further improve the resilience against malicious clients. *RPNet* increases more than $10\times$ query numbers compared to prior defense methods, which increases the attack difficulties. (iii) We provide theoretical analysis on *RPNet* and our implementation of *RPNet* is able to decrease attack success rate by $\sim 91.88\%$ against malicious clients on various neural networks and datasets.

## 2 BACKGROUND AND RELATED WORKS

**Non-interactive Private Inference.** One popular private inference paradigm is based on fully homomorphic encryption (FHE) (Gentry, 2009), which stands out as FHE-based private inference

allows non-interactive neural network inference privacy-preserving, i.e. inference on encrypted data without needing to decrypt it. Compared to interactive private inference based on multi-party computation Mishra et al. (2020), FHE-based private inference has two main advantages: (1) non-interactive end-to-end inference and (2) more secure against layer-by-layer attacks, e.g., client-malicious model extraction attacks. In particular, non-interactive inference is more friendly to clients who have no powerful machines or high-bandwidth network connections. This is because interactive private inference has the drawback that the clients must stay online during the computation. We noticed that many protocols of private inference in Table 1 such as Aharoni et al. (2020); Lehmkuhl et al. (2021), share a similar workflow where the client sends encrypted data to a server. The server then transforms a regular neural network (NN) into a Privacy-preserving Network (PNet), which allows inference on encrypted data without decryption. The original real-number convolution in the NN is replaced with a fixed-point FHE convolution, and the non-linear $ReLU$ function is swapped for polynomial-approximated functions like the $square$ function. The inference result remains encrypted and can only be decrypted by the data owner with a private key, ensuring privacy preservation.

**Black-box Inference Manipulation Attacks.** The malicious-client attack operates on the principle that the attacker can access the black-box MLaaS based on private inference, thereby gaining the capability to manipulate input data to achieve unauthorized access or influence decisions derived from private inference. Current black-box inference manipulation attacks (Guo et al., 2019; Andriushchenko et al., 2020), have been validated as effective in producing adversarial examples that can control the inference output without the necessity for retraining a surrogate model. Specifically, these methods illustrate how a mathematical tool - the discrete cosine transform (DCT), further detailed in Appendix - can be employed to shift an image from the spatial domain to the frequency domain. By initiating a search from lower frequencies and progressing to higher ones, one can effectively pinpoint an adversarial sample, thereby reducing the number of required queries. Our *PNet-Attack* strategy improves the efficiency of attacks by minimizing the number of model queries through a distinct search order and scheduler.

**Resilient Neural Networks against Adversarial-example Attacking Clients.** To defend against query-based black-box attacks, Salman et al. (2020); Byun et al. (2021) show that adding random noise into the input (Qin et al., 2021) or model (Byun et al., 2021) can defend against attacks without perceiving the inputs. Also, R&P (Xie et al., 2017) proposes an input random-transform defense method. RSE (Liu et al., 2018) adds large Gaussian noise into both input and activation and uses ensembles to avoid an accuracy decrease. PNI (He et al., 2019; Cohen et al., 2019; Salman et al., 2019) incorporate noise in the training. However, these defense methods sacrifice enormous accuracy. And the input-transform function in R&P and the ensemble method in RSE introduce a large overhead for PNet. Rusak et al. (2020) introduces that the model with Gaussian augmentation training could defend the common corruptions. RND (Qin et al., 2021) extends the methods in (Rusak et al., 2020; Byun et al., 2021) and achieves the state-of-the-art defense against black-box attacks. However, RND does not consider the distinct features of PNet, i.e., quantized activation and model, and polynomial activation that has a significant decay on the added noise of the input, thus restricting the defense effect on private inference.

**Limitations of Existing Attacks and Defenses on PNet.** Existing black-box attacks and query-based defenses for Neural Network (NN) are not transformed well to PNet. Specifically, we use Figure 2(c) to show that one popular attack SimBA-DCT (Guo et al., 2019) attains $\sim 80\%$ fewer attack success rates on PNet than NN for target attack. This motivates us to design PNet-Attack to identify what adversarial examples are more vulnerable to PNet and how to generate them. Similarly, the encrypted input and additional encoding of PNet make the defense difficult. First, the encrypted input requires a black-box input agnostic defense which has not been well-studied. Second, the polynomial activation, i.e., degree-2 $square$ function, induces a decay effect on the added Gaussian noise of RND method especially when the absolute value of added noise is less than 1. We use Figure 2(d) to show that compared to RND-defense in NN, RND in PNet achieves $\sim 32\%$ lower defense success rate (attack failure rate). This motivates us to design a robust PNet, RPNet, against adversarial attacks.

## 3 PNET-ATTACK: ATTACKS ON SEMI-HONEST PRIVATE INFERENCE

**Attack Threat Model and Use Case.** Our attack strategies are designed to manipulate the output of private inference to gain unauthorized access, potentially resulting in personal or financial benefits,

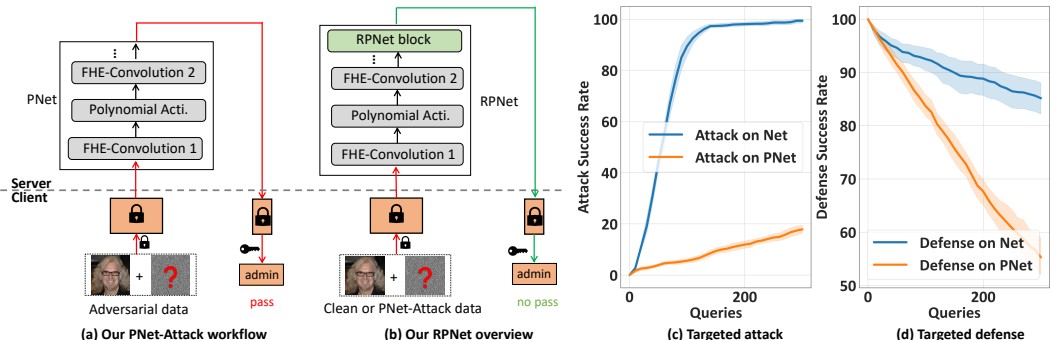

Figure 2: (a) PNet-Attack use case on the privacy-preserving face recognition system. (b) Our RPNet enables a fast, accurate, and robust PNet (c) and (d) Existing black-box attacks and query-based defenses for regular NN are not transformed well to PNet.

as seen in intrusion detection systems (Wyze, 2022; Kuna) and fraud detection systems (Raghavan & Gayar, 2019). These attacks target semi-honest private inference protocols that possess a specific characteristic: the client's final output should coincide with the plaintext output of the final linear layer. Several private inferences, including all the current FHE-based methods (Brutzkus et al., 2019; Chou et al., 2018; Hesamifard et al., 2019; Dathathri et al., 2019; Lou & Jiang, 2021; Benaissa et al., 2021; Aharoni et al., 2020; Lehmkuhl et al., 2021), satisfy this condition. In Figure 2(a), we take a privacy-preserving face-recognition authentication system as an example to illustrate more about our threat model use case, where a non-admin client attacker aims to obtain admin permission. To achieve this goal, the attacker has a motivation to find an adversarial noise or example that causes the system to produce a target-class prediction, e.g., admin, in order to bypass the recognition system. We use Figure 2(b) to show our RPNet can defend against this attack.

**Attack Strategy.** To efficiently manipulate the inference, we propose a PNet-Attack method that is optimized for PNet shown in Algorithm 1. The PNet-Attack method takes one clean image $X \in \mathbb{R}^{d \times d \times c}$, true label $y$, and step size $\epsilon$ as inputs, and generates adversarial perturbation $\delta$, where $d$ is the input width or height, $c$ is channel number. We define the prediction score probability of PNet model as $\mathbf{O} = M_p(x)$. Instead of adding perturbation in the spatial domain, we adopt a more efficient search direction $Q$ in the frequency domain by the existing DCT tool and convert the frequency-domain perturbation $\hat{\alpha}_t \cdot Q$ back to the spatial domain by inverse DCT (IDCT). DCT and IDCT are defined in Appendix. The key idea of the algorithm is simple, i.e., for any direction $Q$ and step size $\epsilon$, one of $x + IDCT(\lambda_t \cdot \epsilon \cdot Q)$ or $x + IDCT(-\lambda_t \cdot \epsilon \cdot Q)$ may decrease $\mathbf{O} = M_p(x)$. We iteratively pick direction basis $Q$ in the ascending order of frequency value $x_{\hat{i},j}$ in $\hat{x}$. Note that we randomly sample one $x_{\hat{i},j}$ when there are multiple entries with an equal value.

The search efficiency of PNet-Attack algorithm is mainly dependent on two components, i.e., arc-shaped search order $Q$ in the frequency domain and perturbation size schedule $\lambda_t$. In particular, frequency-domain input $\hat{x}$ is calculated by $DCT(x)$ for each channel, where the top-left positions of $\hat{x}$ have lower frequency values. Since low-frequency subspace adversarial directions have a much higher density than high-frequency directions, we try to perform the search from lower frequency to higher frequency before a successful attack. To achieve this goal, we iteratively extract the value $x_{\hat{i},j}$ with the lowest frequency from $\hat{x}$. To avoid the repeating search, we pop out the $x_{\hat{i},j}$ from the remaining search space $\hat{x}$ by $\hat{x} = \hat{x}.pop(x)$ shown in Algorithm 1. The search direction basis $Q$ is set as $x_{\hat{i},j}$ for the $t$-th query, which means that we only add the perturbation in the position of $x_{\hat{i},j}$ and check if it decreases the prediction probability at the $t$-th query.

$$\lambda_t = \lambda_{min} + \frac{1}{2}(\lambda_{max} - \lambda_{min})(1 + cos(\frac{t}{T} \cdot \pi)) \quad (1)$$

Since $x_{\hat{i},j}$ with lower frequency may contain more dense information than high-frequency values, we propose a perturbation size schedule $\lambda_t$ to assign larger perturbation size to the positions with lower frequency, which further improves the search efficiency. For $t$-th query, the perturbation size $\alpha_t$ is defined as the multiplication between $\lambda_t$ and frequency-domain perturbation seed $\epsilon$. We define the cosine annealing schedule $\lambda_t$ in Equation 1, where $\lambda_{min}$ and $\lambda_{max}$ are the minimum and maximum coefficients of perturbation size, respectively, and $\lambda_t \in [\lambda_{min}, \lambda_{max}]$, $T$ is the query range cycle.

## 4  RPNET: PRIVATE INFERENCE RESILIENT TO MALICIOUS CLIENTS

**RPNet Design Principle.** In contrast to NN used for plaintext inference, the private inference-designed PNet exhibits unique characteristics, such as polynomial activation and quantized parameters. Consequently, directly adapting previous work (Qin et al., 2021)—which injects Gaussian noise into the input layer—may not be optimal for PNet. We noticed that previous work (Qin et al., 2021) displays a diminished defense impact when applied to private inference. This can be attributed to the significant attenuation of the injected noise by the multi-layer polynomial-approximated activation in PNet, such as the $square$ function, rendering the defense noise virtually ineffective on the model output, especially for deeper neural networks. The diminished noise cannot significantly influence the final-layer logits, thus reducing the defense performance of prior work on PNet. In *RPNet*, we propose a simple, efficient, and provably effective method that circumvents

---

**Algorithm 1** PNet-Attack in Pseudocode

1: **Input:** image $x \in \mathbb{R}^{d \times d \times c}$, label $y$, step size $\epsilon$.
2: adversarial perturbation $\delta = 0$
3: $\mathbf{O} = M_p(x)$, $t = 0$
4: $\hat{x} = DCT(x)$   # for each channel
5: **while** $\mathbf{O}_y = max_{y'} \mathbf{O}_{y'}$ $and$ $t < d^2$ **do**
6:     get $\hat{x_{i,j}}$ with the lowest frequency from $\hat{x}$.
7:     $\hat{x} = \hat{x}.pop(\hat{x_{i,j}})$
8:     $Q = Basis(\hat{x_{i,j}})$
9:     **for** $\hat{\alpha_t} \in \{\lambda_t \cdot \epsilon, -\lambda_t \cdot \epsilon\}$ **do**
10:        $t + +$
11:        $\mathbf{O}' = M_p(x + \delta_t + IDCT(\hat{\alpha_t} \cdot Q))$
12:        **if** $sign(O'_y - O_y) < 0$ **then**
13:            $\delta_{t+1} = \delta_t + IDCT(\hat{\alpha_t} \cdot Q)$
14:            $\mathbf{O} = \mathbf{O}'$
15:            **break**
16: **return** $\delta$

---

the noise decay issue by adding noise to the output layer. Nonetheless, this strategy might be susceptible to an *average inference attack*—a phenomenon we detail in Appendix—particularly when the injected noise has a zero mean. To counteract this, one can simply use non-zero-mean noise and integrate it into the final two layers of the network. To further augment the effectiveness of *RPNet*, we also introduce a novel dynamic noise training (DNT) technique.

In query-based inference manipulation attacks, the aggressor iteratively introduces a minor disruption to the input, subsequently inspecting whether consecutive queries yield varying prediction probabilities. If the probability of the objective prediction for the $(t + 1)$-th query diminishes compared to the $t$-th query, the introduced perturbation is retained. Conversely, the attacker removes the adversarial disruption. The efficiency of the search heavily relies on the correct determination of the perturbation search direction, i.e., whether to add or subtract the disruption in each query. Consequently, a defender can reduce the efficiency of the attack by perturbing the perturbation search direction, thus misleading the prediction probabilities. Inspired by these observations, we propose a swift and accurate defense methodology that simply introduces noise to the output probability in each query, resulting in a robust RPNet. Our RPNet defense approach is designed to achieve two objectives: (i) ensuring that the introduced defense noise doesn't significantly alter the prediction probability of the normal dataset, thus maintaining accuracy, and (ii) ensuring that the introduced defense noise significantly disrupts the attack search direction, thereby reducing the search efficiency.

**RPNet Defense Formulation.** We use Equation 2 to define the prediction probability difference of two queries on PNet, $M_p(x + \delta_t + \mu_t)$ and $M_p(x + \delta_t)$, where $\delta_t$ is the accumulated perturbation at $t$-th query, $\mu_t$ is the perturbation of $t$-th query, e.g., $IDCT(\hat{\alpha_t} \cdot Q)$ if defending PNet-Attack in Algorithm 1.

$$A_p(x, t) = M_p(x + \delta_t + \mu_t) - M_p(x + \delta_t) \tag{2}$$

In Equation 3, we define the main step of the proposed RPNet defense. Two noises $\sigma\Delta_{t+1}$ and $\sigma\Delta_t$ are added into $(t + 1)$-th query and $t$-th query, respectively, to disturb the attack search direction. Those noises are sampled from the same standard Gaussian distribution $\Delta \sim \mathcal{N}(0, 1)$ and multiplied by a small factor $\sigma$. Note that the added noise shares the same encoding method with PNet for correct decryption of prediction result. The key idea of adding noise in the query result is to disturb the difference, i.e., $A_p(x, t)$, of two attack queries and mislead the search directions.

$$
\begin{aligned}
D_p(x, t) &= (M_p(x + \delta_t + \mu_t) + \sigma\Delta_{t+1}) - (M_p(x + \delta_t) + \sigma\Delta_t) \\
&= A_p(x, t) + \sigma(\Delta_{t+1} - \Delta_t)
\end{aligned}
\tag{3}
$$

Specifically, the disturbance success happens when the signs of $A_p(x, t)$ and $D_p(x, t)$ are different. We use Equation 4 to define the probability of disturbance success (DSP). A higher DSP will induce

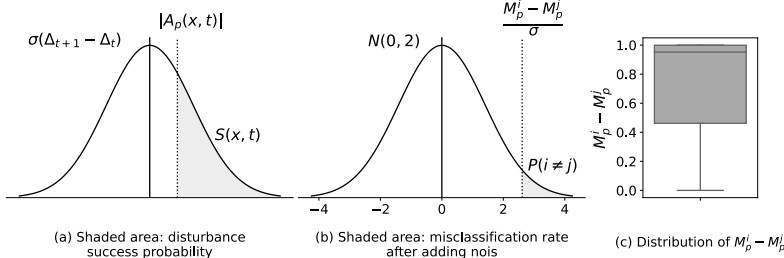

Figure 3: (a) Disturbance success probability S(x, t) for one query. (b) incorrect prediction rate $P(i \neq j)$, which means the probability of misclassification after adding noise on the confidence scores. (c) distribution of the difference between the highest predicted score $M_p^i$ and the second highest predicted score $M_p^j$.

a lower attack success rate (ASR). Therefore, it is of great importance to understand the factors impacting the DSP.

$$S(x, t) = P(sign(A_p(x, t)) \neq sign(D_p(x, t))) \tag{4}$$

We theoretically analyze and calculate the DSP in Equation 5. According to Equation 3, the only difference of $D_p(x, t)$ and $A_p(x, t)$ is $\sigma(\Delta_{t+1} - \Delta_t)$, thus $S(x, t)$ is equal to the probability of adding $\sigma(\Delta_{t+1} - \Delta_t)$ to change the sign of $A_p(x, t)$. Given the Gaussian distribution $\sigma(\Delta_{t+1} - \Delta_t) \sim \mathcal{N}(0, 2\sigma^2)$, the $S(x, t)$ is equal to $1 - \phi(|A_p(x, t)|; 0, 2\sigma^2)$, where $\phi()$ is the cumulative distribution function (CDF) of Gaussian distribution. This is because if $A_p(x, t) < 0$, the added noise sampled from $\mathcal{N}(0, 2\sigma^2)$ should be larger than $|A_p(x, t)|$ to change the sign of $A_p(x, t)$, thus its probability is $1 - \phi(|A_p(x, t)|; 0, 2\sigma^2)$; otherwise, the added noise should larger than $|A_p(x, t)|$ to change the sign of $A_p(x, t)$, thus the probability is also $1 - \phi(|A_p(x, t)|; 0, 2\sigma^2)$. Therefore, using the CDF equation, one can calculate the DSP in Equation 5, where $erf$ is Gauss error function. We demonstrate in Equation 5 that DSP is impacted by two factors, i.e., $|A_p(x, t)|$ and $\sigma$. DSP has a positive relationship with $\sigma$ but is negatively relative to $|A_p(x, t)|$. In Figure 3 (a), we use the shaded area to illustrate the value of $S(x, t)$.

$$\begin{aligned} S(x, t) &= 1 - \phi(|A_p(x, t)|; 0, 2\sigma^2) \\ &= \frac{1}{2} - \frac{1}{2} erf(\frac{|A_p(x, t)|}{2\sigma}) \end{aligned} \tag{5}$$

**RPNet Analysis.** We theoretically analyze the effects of our RPNet defense method on clean accuracy. When applying our PNet on a $n$-class classification task, we can define prediction score as $\mathbf{O} = \{M_p^0, M_p^1, ..., M_p^{n-1}\}$ for clean data. Since our defense method adds Gaussian noise $\sigma\Delta_t$ to the $\mathbf{O}$, we define the prediction score after our defense method as $\mathbf{O}^\sigma = \{M_p^0 + \sigma\Delta_t^0, M_p^1 + \sigma\Delta_t^1, ..., M_p^{n-1} + \sigma\Delta_t^{n-1}\}$. The classification results of $\mathbf{O}$ and $\mathbf{O}^\sigma$ are $i = argmax(O)$ and $j = argmax(O^\sigma)$, respectively. Therefore, RPNet will predict an incorrect classification if $i \neq j$. We use Equation 6 to describe the probability of $P(i \neq j)$ that is positively relative to $\sigma$ but negatively relative to $M_p^i - M_p^j$. In Figure 3 (b), we use the shaded area to illustrate the value of $P(i \neq j)$. Our RPNet achieves a tiny $P(i \neq j)$ and a large $S(x, t)$ given a small $\sigma$, therefore obtaining an accurate and robust PNet. Figure 3 (c) demonstrates the distribution of $M_p^i - M_p^j$ and most of the values are larger than 0.5 on CIFAR-10. The $\frac{M_p^i - M_p^j}{\sigma} > 5$ since the $\sigma$ value is $< 0.1$. Those observations show that $P(i \neq j)$ is tiny since $1 - \phi(\frac{M_p^i - M_p^j}{\sigma} > 5; 0, \sigma^2 = 2)$ is near zero. Therefore, the defender can adjust $\sigma$ based on the validation accuracy and misclassification budget obtained in the training to achieve better defense effectiveness according to Equation 6.

$$\begin{aligned} P(i \neq j) &= P((M_p^i + \sigma\Delta_t^i) < (M_p^j + \sigma\Delta_t^j)) \\ &= \frac{1}{2} - \frac{1}{2} erf(\frac{M_p^i - M_p^j}{2\sigma}) \end{aligned} \tag{6}$$

**RPNet with dynamic noise training (DNT).** By analyzing the Disturbance success probability $S(x, t)$ in Equation 5 and clean accuracy decrease rate $P(i \neq j)$ after applying our RPNet defense method, we reveal that a larger $\sigma$ will improve the defense effect but also may decrease the clean accuracy. To further avoid the clean accuracy decrease, one can reduce the noise sensitivity of PNet model or enlarge the difference between $M_P^i$ and $M_P^j$ in Equation 6. Inspired by those observations, we additionally equip our RPNet with dynamic noise training,

---

**Algorithm 2** RPNet with DNT
- 1: **Input:** RPNet model $M_p$, training data $(x, y)$.
- 2: $t = 0$
- 3: **for** $i = 1$ to $epochs$ **do**
- 4:     Randomly sample $\sigma_i \in [0, \sigma_{max}]$
- 5:     **for** $j = 1$ to $iterations$ **do**
- 6:         $pred = M_p(x + \sigma_i \Delta_t)$
- 7:         $t++$
- 8:         minimize $loss(pred, y)$
- 9:         update $M_p$
- 10: **return** $M_p$

---

denoted by RPNet-DNT, to enable a better balance between clean accuracy and defense effects. We use Algorithm 2 to describe RPNet-DNT that adds dynamic epoch-wise Gaussian noise $\sigma_i \Delta_t$ during each training iteration. Our results in Table 3 show RPNet-DNT attains higher clean accuracy over RPNet.

## 5 EXPERIMENTAL SETUP

**Datasets and Models.** Aligned with recent non-interactive private inference studies Lou & Jiang (2021); Dathathri et al. (2019); Aharoni et al. (2020), we perform our experiments on MNIST (LeCun et al., 2010), CIFAR-10 (Krizhevsky et al., 2014), and Diabetic Retinopathy (Gulshan et al., 2016). For MNIST, we adopt a network with a convolution block and two fully connected layers, as described in HeLayers (Aharoni et al., 2020). For other datasets, we use a structure with three convolution blocks and two fully connected layers. Networks are quantized into 8 bits for MNIST, 10 bits for CIFAR-10, and 16 bits for medical datasets.

**Evaluation Metrics.** Attack Success Rate (ASR): This is the proportion of successful attacks out of the total evaluated images. Higher ASR signifies better attack performance. Average Queries: This represents the mean number of queries for each evaluated image, calculated by dividing the total number of queries by the total number of images. Fewer average queries suggest a more efficient attack. Average $\ell_2$ Norm: This is the mean $\ell_2$ norm for each adversarial image, derived by dividing the total $\ell_2$ norm by the total number of evaluated images. A lower average $\ell_2$ norm indicates a smaller adversarial perturbation. Defense Success Rate (DSR): This is the proportion of successful defenses, equivalent to the attack failure rate. A higher DSR signifies superior defense effectiveness. Clean Accuracy (ACC): This measures the model's accuracy on the non-adversarial (clean) data. Disturbance Success Probability (DSP): This is the likelihood of successfully disrupting the attack search direction. More implementation details are in Appendix.

### 5.1 EXPERIMENTAL RESULTS

**PNet-Attack Evaluation.** In Table 2, the performance of our proposed PNet-Attack is compared against that of SimBA-DCT and the Square attack on CIFAR-10 and a medical dataset. For CIFAR-10, the Square attack achieved a targeted attack success rate (ASR) of 71.56%, requiring an average of 301.6 queries and an average $\ell_2$ norm of 5.44. SimBA-DCT, on the other hand, achieved a slightly higher ASR of 73.98% with a smaller adversarial size, as indicated by a lower $\ell_2$ norm of 4.81. In comparison, our PNet-Attack, even without a perturbation size schedule, improved the ASR by 7.5% while reducing the average $\ell_2$ perturbation norm by 0.72 compared to SimBA-DCT. The introduction of a perturbation size schedule further improved the ASR of PNet-Attack by 12.74%, achieving an average perturbation $\ell_2$ norm of 4.87 with only 201.5 average queries. This is a 22.66% ASR improvement, a 0.61 reduction in average $\ell_2$ norm, and a reduction of 100.1 queries compared to the Square attack. Compared to SimBA-DCT, PNet-Attack with a schedule increased the ASR by 20.24% and reduced the average query count by 100.9, while maintaining a similar average $\ell_2$ norm. For untargeted attacks, PNet-Attack with a schedule improved the ASR by 8.74% and 16.1% over the Square attack and SimBA-DCT, respectively. A similar trend was observed with the PNet-Attack on the medical Diabetic Retinopathy dataset.

Figure 4 (a) and (b) illustrate the attack processes of our PNet-Attack and previous methods on both the CIFAR-10 and Diabetic Retinopathy datasets. With a similar number of queries, our PNet-Attack

Table 2: The attack comparisons of PNet-attack and prior works, e.g., SimBA-DCT (Guo et al., 2019) and Square attack (Andriushchenko et al., 2020) on CIFAR-10 and medical dataset. Untar., Target means untarget and target attacks, respectively.

| Schemes | CIFAR-10 | | | | | | Diabetic Retinopathy | | | | | |
|---|---|---|---|---|---|---|---|---|---|---|---|---|
| | Average Queries | | Average $\ell_2$ | | Success Rate | | Average Queries | | Average $\ell_2$ | | Success Rate | |
| | Untar. | Target | Untar. | Target | Untar. | Target | Untar. | Target | Untar. | Target | Untar. | Target |
| Square | 100.1 | 301.6 | 4.21 | 5.44 | 85.64% | 71.56% | 50.0 | 99.1 | 1.47 | 1.18 | 64.32% | 64.14% |
| SimBA-DCT | 101.6 | 302.4 | 2.86 | 4.81 | 78.28% | 73.98% | 51.8 | 101.0 | 0.82 | 0.92 | 73.28% | 51.49% |
| PNet-Attack | 103.4 | 299.4 | 2.79 | 4.09 | 81.33% | 81.48% | 50.5 | 102.5 | 0.84 | 0.87 | 84.36% | 63.28% |
| +Schedule | 99.8 | 201.5 | 3.61 | 4.87 | **94.38%** | **94.22%** | 50.4 | 98.5 | 1.15 | 1.31 | **89.92%** | **76.48%** |

with scheduling consistently achieves a higher targeted ASR compared to SimBA-DCT and Square attack. This improvement can be attributed to the enhanced attack search efficiency realized through the perturbation scheduling and arc-shaped search order of the PNet-Attack. Additionally, when the adversarial examples have the same $\ell_2$ norm, PNet-Attack still manages to reach a higher ASR compared to other techniques. Specifically, on the CIFAR-10 dataset, PNet-Attack with scheduling secures an ASR over 20% higher than other methods while maintaining an average $\ell_2$ norm of approximately 3.0.

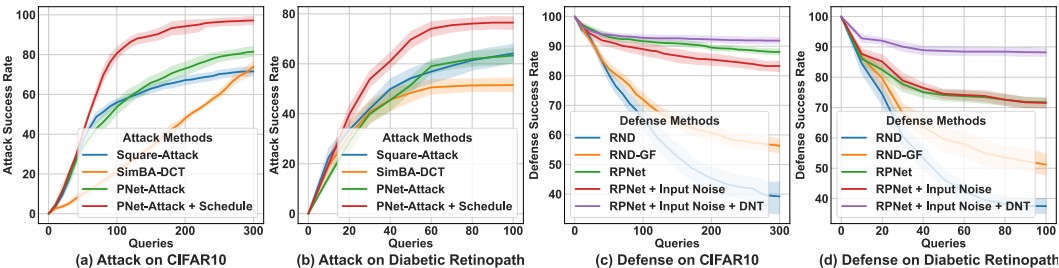

Figure 4: (a, b) Attack success rate v.s. number of queries for different methods. (c, d) Defense success rate v.s. number of queries for RPNet techniques and previous works.

**RPNet Defense Evaluation.** Table 3 presents a comparison of the defense effects achieved by our RPNet and previous methods, including RND and RND-GF, as proposed by (Qin et al., 2021). For targeted attacks on CIFAR-10, RND achieves a Defense Success Rate (DSR) of 39.22% while maintaining a clean accuracy of 72.86%. The RND-GF method, which includes Gaussian noise during training, reaches a defense success rate of 56.33% with an accuracy of 73.71%. In contrast to RND which incorporates noise into the input, our RPNet method introduces noise into the logits, which notably enhances the DSR by approximately 30%. This improvement can be attributed to the fact that adding noise to the input of PNet with a polynomial activation function considerably diminishes the noise. However, introducing noise to the output bypasses this decay, as substantiated by our theoretical analysis of RPNet and empirical results. For example, RPNet-DNT registers a DSR of 91.88% while maintaining a higher clean accuracy of 74.55%. When compared to RND-GF in the context of targeted attacks on CIFAR-10, RPNet-DNT enhances DSR by 35.55% and clean accuracy by 0.84%.

Our RPNet and RPNet-DNT demonstrate consistent enhancements during untargeted attacks and across other medical datasets. Specifically, RPNet-DNT shows a notable increase in untargeted defense success rate by 52.97% compared to RND-GF on CIFAR-10. Likewise, on the Diabetic Retinopathy dataset, RPNet exhibits a rise of 14.54% and 39.69% in the targeted and untargeted defense success rates, respectively, over RND-GF, while achieving a 0.18% increase in clean accuracy. The injection of noise into the input doesn't yield significant improvements in defense success rates. However, the DNT technique notably enhances both the defense success rate and clean accuracy.

Figure 4 (c) and (d) present the defense outcomes of prior works RND, RND-GF, and our techniques, including RPNet, RPNet+Input noise, and RPNet-DNT with input noise. Notably, all techniques display a high defense success rate in the initial queries due to the low attack success rate associated with a limited number of queries. However, as the number of queries increases, the defense success

Table 3: The defense comparisons of RPNet and prior works, e.g., RND (Qin et al., 2021), RND-GF (Qin et al., 2021) and Adversarial Training(AT) (Goodfellow et al., 2014), on CIFAR-10 and medical dataset. '+Input noise', and '+DNT' represent adding Gaussian noise into the input layer and using an additional DNT method, respectively, on RPNet.

| Schemes | CIFAR-10 | | | | | | Diabetic Retinopathy | | | | | |
|---|---|---|---|---|---|---|---|---|---|---|---|---|
| | Clean Accuracy | | Average queries | | Success Rate | | Clean Accuracy | | Average queries | | Success Rate | |
| | Untar. | Target | Untar. | Target | Untar. | Target | Untar. | Target | Untar. | Target | Untar. | Target |
| RND | 72.86% | | 199.8 | 301.1 | 2.03% | 39.22% | 66.81% | | 48.5 | 53.1 | 15.00% | 46.40% |
| RND-GF | 73.71% | | 204.2 | 300.4 | 10.39% | 56.33% | 67.73% | | 50.2 | 49.3 | 10.08% | 59.60% |
| AT | 67.88% | | 199.5 | 302.5 | 49.08% | 86.20% | 61.37% | | 50.0 | 51.3 | 37.36% | 79.05% |
| RPNet | 74.10% | | 198.2 | 299.1 | 56.17% | 88.04% | 67.91% | | 51.7 | 50.1 | 49.77% | 74.14% |
| +Input noise | 73.53% | | 202.7 | 300.1 | 49.69% | 83.28% | 65.82% | | 49.3 | 50.7 | 36.17% | 74.53% |
| +DNT | **74.55%** | | 199.4 | 299.7 | **63.36%** | **91.88%** | 68.09% | | 48.9 | 52.0 | **66.41%** | **88.67%** |

rates of both RND and RND-GF significantly decline. In contrast, our RPNet maintains a high defense success rate. Similar to RND-GF, the addition of noise to the input layer in RPNet+Input noise doesn't result in a significant improvement in defense. This suggests that the input noise might be diminished by the polynomial activation of PNet. With the incorporation of DNT techniques, RPNet-DNT further enhances the defense outcomes. It's important to note that without adding noise to the output layer, RPNet with input noise and DNT still fails to sustain a high defense effect.

Figure 5 (a) and (b) highlight the superior defense efficiency of RPNet in comparison to RND. By achieving a higher Disturbance Success Probability in both targeted and untargeted attacks, RPNet provides an empirical explanation for its heightened defense efficacy. Figure 5 (c) and (d) demonstrate that RPNet strikes a more effective balance between defense effect and accuracy than RND-GF. Specifically, for a given $\sigma$ (e.g., 0.1), Figure 5 (c) shows that RPNet realizes a higher defense success rate than RND-GF. Similarly, Figure 5 (d) indicates that, for the same $\sigma$, RPNet achieves a higher clean accuracy. RPNet therefore exhibits less noise sensitivity than RND-GF.

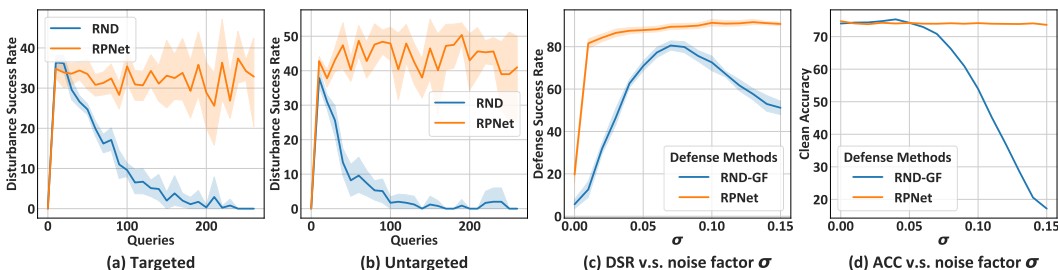

Figure 5: (a, b) RPNet achieves a higher probability of disturbance success probability than prior work RND. (c, d) RPNet achieves a better balance between defense effect and clean accuracy.

**Ablation Study.** As depicted in Figure 5 (c) and (d), RPNet outperforms RND-GF in terms of striking a superior balance between defense proficiency and accuracy. Specifically, for a given $\sigma$ value (e.g., 0.1), Figure 5 (c) reveals that RPNet yields a higher defense success rate compared to RND-GF. Meanwhile, Figure 5 (d) indicates that, for an equivalent $\sigma$ value, RPNet secures higher clean accuracy. Consequently, RPNet demonstrates less sensitivity to noise than RND-GF.

## 6 CONCLUSION

This work first introduces PNet-Attack, an innovative inference manipulation attack for private inference protocols reliant on fully homomorphic encryption. The attack necessitates $3\times \sim 8\times$ fewer queries than current approaches. Moreover, we present RPNet, a robust private inference protocol designed to withstand malicious clients, achieved by incorporating cryptography-compatible noise in the feature and logits layers and deploying a DNT technique. RPNet demands over $10\times$ more query numbers compared to previous defense methods, substantially elevating the attack difficulty. Theoretical analysis and empirical testing show that RPNet can diminish the attack success rate by approximately $91.9\%$ across various neural networks and datasets.

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
