# OpenReview forum: "RPNet: Robust Non-Interactive Private Inference against Malicious Clients with Adversarial Attacks"
_ICLR.cc/2024/Conference — ICLR 2024 Conference Withdrawn Submission_

### Official Review · Reviewer_PzFf · 2023-10-24

**Soundness:** 2 fair
**Presentation:** 1 poor
**Contribution:** 2 fair
**Rating:** 3
**Confidence:** 3

**Summary:**

This paper provides a means of finding adversarial examples for a neural network being provided in a machine learning as a service. It also provides a way of noising responses from the network so as to impede this attack.

**Strengths:**

The attacks seems pretty effective compared to the state of the art.
The defence massively reduces the effectiveness of the attack at very little cost to accuracy.

**Weaknesses:**

The attack requires access to the last layer of the network rather than just an argmax of it. This would be an unusual thing to release in a lot of situations.

The defence is tailored specifically to their attack. In general it is easy to defend against a chosen attack, this is very little evidence that the defence(s) would hold up against an intelligent attacker tailoring an attack to them.

There are a few definitions that really aren't. For instance the paper says they define prediction score probability to be given by O=M_p(x). But this M_p(x) expression has not been defined previously. Perhaps they mean that M_p(x) is defined to be the prediction score probability but if so the sentence is backwards and it still doesn't work as I don't know what "prediction score probability" means. The authors should be careful about defining things before they use them.

The paper talks a lot to begin with about using multiparty computation to operate the Machine Learning as a Service (MLaaS) and how it isn't enough for it to be semi-honestly secure because of attacks like this. This makes little sense, a maliciously secure protocol that outputted the last layer's values would be vulnerable to this attack and a semi-honestly secure protocol that didn't output the last layer wouldn't be (though it might be vulnerable to other attacks as a result of not being maliciously secure). The attack in this paper even makes sense is the MLaaS is providing no privacy guarantees to the clients at all but is jsut a server that takes client inputs and evaluates the model on them in the clear.

**Questions:**

Why do you talk so much about semi-honest security?

Is there any reason to think this defence would work against arbitrary attacks?

Why shouldn't the service just hide the values outputted by the last layer behind an argmax function?

---

### Official Review · Reviewer_7dXp · 2023-10-27

**Soundness:** 1 poor
**Presentation:** 3 good
**Contribution:** 1 poor
**Rating:** 3
**Confidence:** 4

**Summary:**

In this paper, the authors explore an interesting problem - adversarial attacks and defenses under secure computation based PPML protocols. They first optimize a search-based method to generate adversarial samples to cause classifications of encrypted ML models. Then they propose a defense (adding noise to the output layer) that can mitigate the diminished noise due to polynomial activation and quantized ML model parameters. The authors also evaluate their attacks and defenses over benchmarking datasets.

**Strengths:**

1. The research question of this paper seems interesting. Considering other attack surfaces in PPML is important.

2. This paper is generally easy to read.

**Weaknesses:**

1. The proposed attack seems to be agnostic to PPML protocols. By nature, PPML attempts to preserve the model's behavior and performance. Namely, encrypted model trivially suffers from adversarial attacks. The insights look limited to me.

2. The proposed defense is not sound because it cannot integrated into the PPML pipeline. Based on my understanding, the defense will add noise on the plaintext output of the model. However, in PPML, the server can't output an plaintext confidence vector and add noises accordingly.

3. The evaluation is not conducted on the PPML protocols. Only quantized models are used, which cannot demonstrate the performance of the attack and defense in a fair manner.

**Questions:**

See the weaknesses above.

Suggestions:

1. Integrate your defense in the pipeline of PPML

2. Evaluate your attack and defense in the PPML protocols, including protocols with/without polynomial approximations, quantized models, and designs using fixed point values/secure truncations for secure computation.

---

### Official Review · Reviewer_gjYP · 2023-11-01

**Soundness:** 3 good
**Presentation:** 3 good
**Contribution:** 2 fair
**Rating:** 5
**Confidence:** 2

**Summary:**

This paper addresses security issues that can arise when performing privacy-preserving machine learning models. The client continuously queries the encrypted image while proposing an attack that adds noise to obtain adversarial noise. This allows them to obtain adversarial images, potentially gaining administrative authority. To mitigate such attacks, the proposed approach involves adding stochastic noise to the artificial intelligence model, disrupting the ability to obtain adversarial noise with each query. Furthermore, to reduce accuracy loss caused by this method, dynamic noise training techniques are employed.

**Strengths:**

This study highlights new security issues that can arise during the process of performing information security artificial intelligence services. Until now, research on information security AI models has only considered security issues regarding client data. However, it has demonstrated the possibility of clients exploiting these points to exceed their authority. Therefore, I consider this research to open up a new perspective in the field of privacy-preserving machine learning.

**Weaknesses:**

Some previous papers have efficiently used the square function as an activation function to create information security AI models. However, this paper criticizes this as a significant departure from conventional AI models and proposes a new attack strategy by improving existing techniques. Nevertheless, I feel that this paper's technology is not qualitatively different from existing methods, and I think its theoretical contribution is somewhat weak. The framework of the attack is similar to existing attack algorithms, and the paper does not sufficiently explain how meaningful the differences between these attack algorithms are in the context of current privacy-preserving AI models. Specifically, on page 4, when explaining the attack algorithm, it does not clarify how it differs from previous attack techniques and what significance those differences hold in the current privacy-preserving AI model context.

**Questions:**

1) On page 4, what is the difference between the proposed attack and the previous attacks? Please explain the reason for this difference.
2) What do the authors think is the main technical novelty in their attack algorithm and defense algorithm compared to the previous algorithms?

---

### Official Review · Reviewer_Uygt · 2023-11-01

**Soundness:** 1 poor
**Presentation:** 1 poor
**Contribution:** 1 poor
**Rating:** 3
**Confidence:** 5

**Summary:**

First, this paper designed an inference manipulation attack against semi-honest private inference protocols. Then this paper implemented RPNet, a private inference protocol that can withstand malicious clients.

**Strengths:**

This paper implemented a private inference protocol that can withstand adversarial attacks.

**Weaknesses:**

1. This paper first designed an inference manipulation attack against semi-honest private inference protocols. I don't think this makes sense. When considering a semi-honest inference protocol, the protocol already limits the adversary's capabilities, that is, it is semi-honest. It is contradictory and unjustifiable to consider malicious adversaries contrary to this restriction. In addition, semi-honest inference protocols are bound to be unable to resist malicious adversaries. This is a consensus and can be easily demonstrated.
2. In the privacy inference, if the client is malicious, then the inference protocol should be able to tolerate any malicious behavior of the client. However, this paper only considers adversarial attacks, which is very narrow. In addition, in the process of privacy inference, various cryptographic technologies are required, such as ZK, malicious MPC, etc. to regulate the behavior of the malicious client. It is not safe to only use FHE in this paper. This paper also lacks the necessary security analysis to support the proposed private inference process.
3. The start-of-the-art and mainstream private inference works utilized HE for linear layers and OT for non-linear layers. This paper is backward in using FHE for the entire inference evaluation.

**Questions:**

See above.